# Characterising within-hospital SARS-CoV-2 transmission events using epidemiological and viral genomic data across two pandemic waves

Benjamin B. Lindsey [1,2,15], Ch. Julián Villabona-Arenas [3,4,15], Finlay Campbell[5,15], Alexander J. Keeley [1,2], Matthew D. Parker [6,7,8], Dhruv R. Shah [1], Helena Parsons[1,2], Peijun Zhang[1], Nishchay Kakkar[2], Marta Gallis[1], Benjamin H. Foulkes [1], Paige Wolverson [1], Stavroula F. Louka[1], Stella Christou[1], Amy State[2], Katie Johnson[2], Mohammad Raza[1,2], Sharon Hsu [1,7], Thibaut Jombart[3,4,9], Anne Cori[9], Sheffield COVID-19 Genomics Group*, The COVID-19 Genomics UK (COG-UK) consortium, CMMID COVID-19 working group*, Cariad M. Evans[1,2], David G. Partridge [1,2], Katherine E. Atkins [3,4,10 ✉], Stéphane Hué [3,4 ✉] & Thushan I. de Silva [1,2,11 ✉]

Hospital outbreaks of COVID19 result in considerable mortality and disruption to healthcare services and yet little is known about transmission within this setting. We characterise within hospital transmission by combining viral genomic and epidemiological data using Bayesian modelling amongst 2181 patients and healthcare workers from a large UK NHS Trust. Transmission events were compared between Wave 1 (1st March to 25th July 2020) and Wave 2 (30th November 2020 to 24th January 2021). We show that staff-to-staff transmissions reduced from 31.6% to 12.9% of all infections. Patient-to-patient transmissions increased from 27.1% to 52.1%. 40%-50% of hospital-onset patient cases resulted in onward transmission compared to 4% of community-acquired cases. Control measures introduced during the pandemic likely reduced transmissions between healthcare workers but were insufficient to prevent increasing numbers of patient-to-patient transmissions. As hospital-acquired cases drive most onward transmission, earlier identification of nosocomial cases will be required to break hospital transmission chains.

[1] The Florey Institute for Host-Pathogen Interactions & Department of Infection, Immunity and Cardiovascular Disease, Medical School, University of Sheffield, Sheffield, UK. [2] Sheffield Teaching Hospitals NHS Foundation Trust, Sheffield, UK. [3] Centre for Mathematical Modelling of Infectious Diseases, London School of Hygiene and Tropical Medicine, London, UK. [4] Department of Infectious Disease Epidemiology, London School of Hygiene and Tropical Medicine, London, UK. [5] Health Emergencies Programme, World Health Organization, Geneva, Switzerland. [6] Sheffield Biomedical Research Centre, The University of Sheffield, Sheffield, UK. [7] Sheffield Bioinformatics Core, The University of Sheffield, Sheffield, UK. [8] The Department of Neuroscience/ Neuroscience Institute, The University of Sheffield, Sheffield, UK. [9] MRC Centre for Global Infectious Disease Analysis, Department of Infectious Disease Epidemiology, School of Public Health, Imperial College London, London, UK. [10] Usher Institute, The University of Edinburgh, Edinburgh, UK. [11] MRC Unit The Gambia at the London School of Hygiene and Tropical Medicine, Fajara, The Gambia. [15] These authors contributed equally: Benjamin B. Lindsey, Ch. Julián Villabona-Arenas, Finlay Campbell. *Lists of authors and their affiliations appears at the end of the paper. ✉email: Katherine.Atkins@ed.ac.uk; Stephane.Hue@lshtm.ac.uk; t.desilva@sheffield.ac.uk

Severe acute respiratory syndrome coronavirus 2 (SARS-CoV-2) has resulted in multiple hospital outbreaks, exposing healthcare workers (HCWs) and non-COVID-19 patients to SARS-CoV-2 infection[1–3]. At least 32,307 patients are thought to have been infected in hospitals in England and Wales and an estimated 414 HCWs died between March and December 2020[4]. This is probably a substantial underestimate as it represents the number of individuals that meet the narrow definitions for healthcare-associated infection rather than the true number of infected individuals in hospitals. To safely continue routine and elective activities in hospitals during times of high SARS-CoV-2 incidence, it is important to discern factors that drive hospital-acquired infections. This greater understanding can be used to protect staff and patients, as well as informing further efforts to contain hospital outbreaks.

Identifying within-hospital SARS-CoV-2 transmission events using epidemiological data remains challenging for two reasons. Firstly, the high variability in the viral incubation period means it is often difficult to determine whether hospital onset cases are community- or hospital-acquired. Secondly, at least 33% of SARS-CoV-2 infections in adults are thought to be asymptomatic[5], therefore identifying all patients and HCWs contributing to transmission is challenging. In the limited instances where viral genomic data were analysed, this information was used to confirm or complement a purely epidemiological approach[6–9]. Elucidating the source of transmission events on the basis of viral genetic relatedness alone also entails considerable uncertainty due to the slow evolutionary rate of SARS-CoV-2[10]. In the time scale of an outbreak, a large proportion of individuals are infected by viruses too genetically similar to each other to distinguish genuine transmission events from unrelated infections. Furthermore, data from HCWs have rarely been included in previous analyses[11] and the relative role that patients and HCWs have played in fuelling hospital outbreaks in the UK remains largely unknown.

The integration of genomic, epidemiological and location data into a statistical inference framework offers a possible route to more accurate estimates of within-hospital transmission. Under such an approach, a transmission event between a pair of individuals is supported if their symptom onset times are compatible with the serial interval distribution SARS-CoV-2, if the individuals are in the same hospital location at the time of a suspected transmission event and if their viral genomes exhibit a high degree of relatedness.

In this study, we reconstructed SARS-CoV-2 outbreaks in a large NHS teaching hospital trust in England during the first two UK epidemic waves. We integrated over 2000 viral genomic sequences, patient and staff locations, and routinely available epidemiological information in a Bayesian framework that incorporates prior knowledge on the relative contributions of within-ward and between-ward transmission, as well as the proportion of cases involved in transmission chains that were not represented in the dataset[12,13]. Using this approach, we characterised the dynamics of SARS-CoV-2 transmission within a hospital setting, identifying key differences across the two pandemic waves, as well as the relative contribution of different groups and hospital locations to within-hospital transmission.

## Results

**Study population, SARS-CoV-2 testing and infection control measures.** During the first wave of the UK epidemic (Wave 1; defined as 1st March to 25th July 2020 for our analysis), 886/1,184 (74.8%) patients and 842/1,104 (76.3%) HCWs at STHNFT who tested positive for SARS-CoV-2 had sequence data available with over 90% genome coverage (Fig. 1). During the second wave

(Wave 2; defined here as 30th November 2020 to 24th January 2021) 669/1183 (56.6%) SARS-CoV-2 positive patients and 651/838 (77.7%) SARS-CoV-2 positive HCWs had sequence data available with over 90% genome coverage. Cases were excluded if they were outpatients, non-clinical staff (not working in clinical areas; e.g. Accounting, Information Technology, Catering), non-STHNFT or community-based staff, household contacts of staff members, and staff who had missing ward location data, leaving 1302 individuals in Wave 1 and 879 individuals in Wave 2 for the analysis (Fig. 1b, Table 1).

SARS-CoV-2 testing policy and infection prevention and control (IPC) measures evolved throughout the pandemic (Fig. 1a). Testing was performed in symptomatic patients with suspected SARS-CoV-2 infection on admission throughout the study period, with testing offered to symptomatic staff from 17th March 2020[14]. Testing of all admissions regardless of symptoms commenced on 25th April 2020 and screening of all asymptomatic patients and staff on wards with outbreaks from 18th May 2020. In addition to screening on admission, all patients were routinely tested on day 5 of admission from 1st September 2020. Routine twice weekly testing using lateral flow devices, followed by confirmatory Nucleic Acid Amplification Test (NAAT), was offered to staff in all clinical areas from 8th December 2020. Level 2 personal protective equipment (PPE; aprons, gloves, eye protection and fluid resistant surgical face mask) was used by staff only for seeing suspected COVID-19 cases from 17th March 2020, and for all patient contact from 8th April 2020. HCWs were mandated to wear surgical face masks in all areas of the hospital from 15th June 2020. The SARS-CoV-2 staff vaccination programme commenced on 10th December 2020.

Viral genomes were classified into 64 different PANGO lineages for Wave 1 and 24 lineages for Wave 2. Lineages B.1.1.1 (471/1,302, 36.2%), B.1.1.119 (180/1,302, 13.8%) and B.1 (110/1,302, 8.4%) predominated during Wave 1, while lineages B.1.177 (293/879, 33.3%), B.1.1.7 (263/879, 29.9%) and B.1.177.4 (112/879, 12.7%) predominated during Wave 2 (Supplementary Fig 1).

**Quantifying hospital-acquired infections.** Using admission dates, symptom onset dates, SARS-CoV-2 positive test dates, ward location of cases and comparison between consensus viral genome sequences, our model inferred likely hospital transmissions between individuals, together with (i) the time of those events, (ii) the ward on which the transmission occurred, and (iii) whether the infector was a sampled case or a case involved in transmission but not represented in the dataset.

From the 1302 cases in our dataset from wave 1, our model identified 388 (95% credible interval (CI) 292–479) hospital-acquired infections, along with 85 (95% CI 21–192) further cases estimated to be involved in transmission but not represented in the dataset (total 473 hospital-acquired infections, 95% CI 310–688). During Wave 2, 350 (95% CI 285–410) cases from the 879 in the dataset were estimated to be hospital-acquired infections, with another 52 (95% CI 11–120) infections estimated that were not represented in the dataset (total 402, 95% CI 293–538). In Wave 1, patient cases comprised 40.9% (95% CI 36.1–45.5%) of all sampled hospital-acquired cases, compared to 65.1% (95% CI 60.3–69.4%) of all sampled hospital-acquired cases in Wave 2. Our model estimates showed good agreement with previous a priori inpatient epidemiological definitions of community or hospital onset categories (Table 1 and Supplementary Table 2)[15]. Specifically, no 'community onset-community associated' patient cases were identified as hospital-acquired by our model during either wave, and the majority of 'hospital onset-hospital acquired' and 'hospital onset-suspected

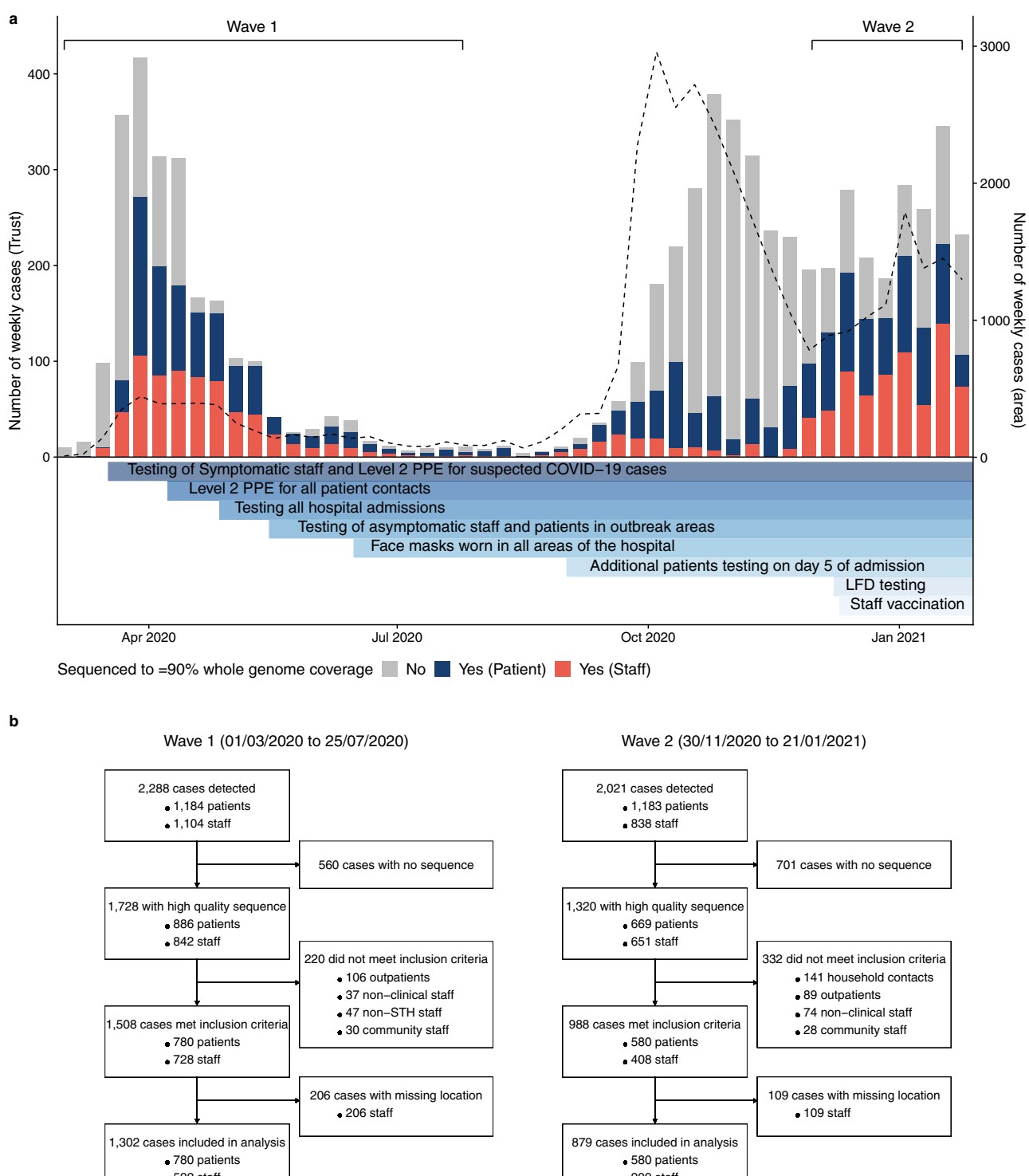

**Fig. 1 SARS-CoV-2 positive staff and healthcare worker samples included in the study. a** SARS-CoV-2 positive cases in patients and staff at Sheffield Teaching Hospitals NHS Foundation Trust (STH) over time (left Y axis), and implementation of testing, prevention and control interventions. All SARS-CoV-2 nucleic acid amplification tests from STH patients and staff found to be positive in the hospital (pillar 1) diagnostic laboratory are shown in grey bars. Sequences with genome coverage of 90% or higher for samples from healthcare workers (red) and patients (blue) are shown. These relate to samples with high-quality sequences (1728 in Wave 1 and 1320 in Wave 2) shown in Fig. 1b. The dashed line shows weekly case numbers in the Sheffield area (right Y axis); data taken from ref. [41]. Level 2 positive protective equipment (PPE)—aprons, gloves, eye protection and fluid resistant surgical face mask. LFD testing —Lateral flow device testing for staff two times per week. Grey bars represent SARS-CoV-2 cases tested (staff and patients) testing positive in the STH pillar 1 diagnostic laboratory with no high-quality sequence available. Wave 1 and Wave 2 denote periods and samples included in the study. **b** Details of SARS-CoV-2 positive cases from patients and healthcare workers sequenced and included in the study.

**Table 1 Summary of study cohort.**

| | Wave 1 | Wave 2 |
|---|---|---|
| Dates | 1st March 2020 to 25th July 2020 | 30th November 2020 to 24th January 2021 |
| Total Cases | 1302 | 879 |
| Patients | 780 (59.9%) | 580 (66.0%) |
| Staff | 522 (40.1%) | 299 (34.0%) |
| Number of hospital locations | 120 | 93 |
| Global PANGO lineages | 64 | 24 |
| Fraction of patients with inferred symptom onset date | 30.1% | 22.0% |
| Median ward movements per patient (min-max) | 2 (1-9) | 2 (1-9) |
| Patient classification | | |
| Community onset—community associated | 486 (62.3%*) | 236 (40.7%*) |
| Community onset—suspected healthcare associated | 79 (10.1%*) | 67 (11.6%*) |
| Hospital onset—healthcare associated | 80 (10.3%*) | 100 (17.2%*) |
| Hospital onset—probable healthcare associated | 74 (9.5%*) | 88 (15.2%*) |
| Hospital onset—intermediate healthcare associated | 61 (7.8%*) | 89 (15.3%*) |

Number of hospital locations includes wards and non-clinical areas. Ward movements refer to between ward movements and do not include bed movements within the same ward. Classification of patient cases according to likely source of infection (community or hospital-acquired) is based on SAGE criteria[15]. Community onset-community associated = positive test up to 14 days before or within 2 days after hospital admission; Community onset-suspected healthcare associated = positive test up to 14 days before or within 2 days after admission, with discharge from hospital within 14 days before test; Hospital onset-intermediate healthcare associated = positive test 3–7 days after hospital admission with no discharge from hospital in 14 days before the specimen date; Hospital onset-suspected healthcare associated = positive test 8–14 days after admission or 3–14 days after admission with discharge from hospital in 14 days before test; Hospital onset-healthcare associated = positive test 15 or more days after hospital admission. Asterix represents percentage of patient cases.

hospital acquired' cases were identified as likely hospital-acquired by our model.

**Transmission chain reconstruction**. We identified 95 (95% CI 82–109) transmission chains (defined as contiguous transmission events between 2 or more cases) in Wave 1 and 72 (95% CI 61–84) transmission chains in Wave 2 (Supplementary Fig. 2). The median number of cases per transmission chain was 3 (95% CI 3– 4) in Wave 1 and 4 (95% CI 3–5) in Wave 2. A staff member was identified as the index case in 50.6% (95% CI 42.0– 58.0%) of transmission chains in Wave 1 and in 31.3% (95% CI 23.1–39.7%) of transmission chains in Wave 2. Forty different PANGO lineages were involved in transmission chains in Wave 1 and 13 were found in Wave 2 chains.

The inferred transmission events entailed considerable uncertainty (Fig. 2a–b) but despite that, a pattern on the nature of the links was found (Fig. 2c). Of the transmissions between sampled cases in Wave 1, 31.6% (104/329, 95% CI 26.9–35.8%) were staff-to-staff events, 27.1% (89/329, 95% CI 23.3–31.4%) were patient-to-patient, 25.5% (84/329, 95% CI 22.1–29.3%) were patient-to-staff and 15.5% (51/329, 95% CI 12.2–19.1%) were staff-to-patient (Fig. 2c). By contrast, during Wave 2, the majority of transmission events (162/311; 52.1%, 95% CI 48.0–57.1%) between sampled cases were patient-to-patient events, with 21.2% (66/311, 95% CI 18.0–24.1%) patient-to-staff, 13.5% (42/311, 95% CI 10.1–17.5%) staff-to-patient and 12.9% (40/311, 95% CI 9.5–15.9%) staff-to-staff transmission events (Fig. 2c).

In Wave 1, 55.3% (104/188, 95% CI 48.9–61.2%) of staff infections resulted from another staff case, which decreased to 37.7% (40/106, 95% CI 29.3–45.4%) in Wave 2. In Wave 1, 63.6% (89/140, 95% CI 56.1–71.0%) of patient infections resulted from another patient case which increased to 79.4% (162/204, 95% CI 73.7–84.6%) in Wave 2.

**Ward and Bay level transmission**. Identified transmission events were not evenly distributed across the 132 hospital locations included but isolated to 38 wards in three of the five hospitals within STHNFT. The eight wards with the highest number of infections in Wave 1 accounted for 51.0% (95% CI 39.7–63.4%) of all transmissions, indicating the presence of transmission hot spots. A similar finding was observed during Wave 2, where 10 wards accounted for 50.1% (95% CI 40.6–60.5%) of all

transmission events (Fig. 3). We found evidence that the relative importance of specific wards in contributing to overall transmission was maintained across the two waves (Spearman's Rank correlation Rho 0.54, $P < 0.0001$, Ranked by mean number of transmissions per ward). However, there was considerable variability between waves, and several wards that were transmission hotspots in Wave 1 did not make up the 10 wards accounting for >50% of transmissions in Wave 2 (Fig. 3a–b). Equally, several wards with no transmission event during Wave 1 were identified as transmission hotspots in Wave 2. Very few transmissions were estimated to have occurred in critical care units (one in Wave 1 and none in Wave 2). Considerable variability was also seen between wards in the proportion of infectors and infectees made up by patients and staff (Fig. 3c–f). The highest number of infections on a single ward was 50 (95% CI 40–58) in Wave 1 but decreased to 30 (95% CI 23–36) in Wave 2. The highest number of separate transmission chains on a single ward was 8 (95% CI 5–11) for Wave 1 and 5 (95% CI 3–7) for Wave 2.

Wards comprise a combination of multi-bed bays with shared bathroom facilities and individual en-suite side rooms. We used a post hoc analysis to evaluate the contribution of bay-level transmission between patients to the outbreak. We identified 38.3% (95% CI 29.9–47.1%) of patient–patient transmissions in Wave 1 and 33.8% (95% CI 27.9–39.6%) in Wave 2 were between patients who shared a bay at some point during their stay. We estimated an increased risk of transmission between individuals who shared a bay compared with those who shared a ward as 2.8 (95% CI 2.2–3.5) times higher in Wave 1 and a 2.5 (95% CI 2.1–2.9) times higher in Wave 2.

**Secondary cases**. The crude mean number of secondary cases was 0.30 (95% CI 0.21–0.38) for Wave 1 and 0.40 (95% CI 0.31–0.48) for Wave 2. Adjusting for cases involved in transmission but not represented in the dataset, we estimated that 51.3% (95% CI 49.6–53.1%) of infections in Wave 1 and 43.6% (95% 42.3–45.0%) of infections in Wave 2 resulted in no onward transmission. Only 0.2% (95% CI 0.04–0.4%) infections in Wave 1 and 0.6% (95% CI 0.4–0.9%) infections in Wave 2 resulted in more than 5 secondary cases (Fig. 4). Fewer patients classified as having community onset-community associated infections gave rise to secondary cases within the hospital (3.7%, 95% CI 1.8–5.6% for Wave 1;

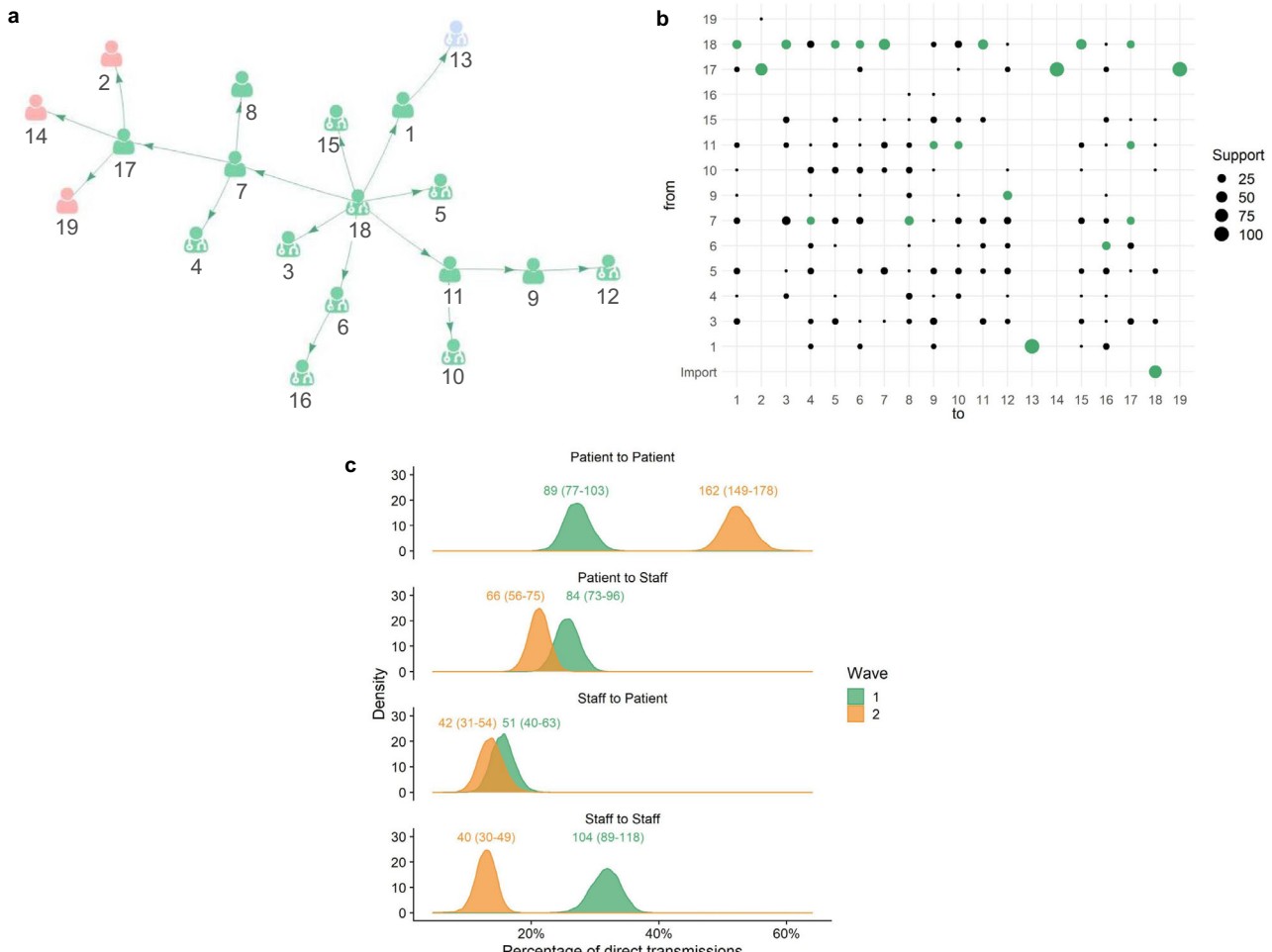

**Fig. 2 Within-hospital transmission chains and estimated infections within and between patients and healthcare workers. a** An example transmission chain showing transmission between staff (icons with stethoscope) and patients (icons without stethoscope). Icons are labelled with their case identification number. Each colour represents a separate ward where the infection occurred. **b** The support (the percentage of networks across all those sampled, where a given infector is assigned to each case) for each potential transmission pair shown in the example network in panel 2a. Numbers on the plot correspond to the case identification numbers in Fig. 2a and the green circles correspond to the transmission pairs displayed. Import = likely community-acquired infection imported into hospital. From = infector, to = infectee. **c** Comparison of the percentage of each transmission type between the two waves. The distributions display the percentages throughout the 10,000 plausible networks. The numbers above each distribution are the absolute numbers of each transmission pair with 95% credible intervals shown within the brackets.

3.5%, 95% CI 1.7–5.9% for Wave 2) compared to those with hospital onset-hospital acquired infections (45.5%, 95% CI 33.8–55.0% for Wave 1; 51.2%, 95% CI 41.0–60.0% for Wave 2), or other categories of hospital-onset cases (Fig. 4b, Supplementary Table 2). All findings were consistent across sensitivity analyses in which we either relaxed the probability of the most recent sampled ancestor $\alpha_i$ and infectee $i$ being registered on the same ward on the day of transmission or changed $\pi$, the proportion of all cases in the outbreak represented in the dataset (ranging from 30 to 70%) (Supplementary Figs. 3–4).

## Discussion

To our knowledge, our findings represent the largest collection of SARS-CoV-2 genomic and hospital epidemiology data to date used to reconstruct directional transmission networks, where we estimated hospital-acquired SARS-CoV-2 infections across two pandemic waves in the UK using a Bayesian framework. Importantly, our model also accounts for events within the identified transmission networks that were not represented in the dataset, which is crucial given the likely presence of unidentified infections or those

lacking sequence data. We observed different contributions to the total number of within-hospital transmission events from those occurring between and within staff and patients across the two waves. We identified transmission hotspots within our institution, with a relatively small proportion of locations accounting for most hospital-acquired infections in staff and patients. We also found that the majority of SARS-CoV-2 infections resulted in onward transmission, with secondary cases identified in >50% of infections but relatively few so called 'superspreader' events.

While much attention has been paid to staff potentially acquiring SARS-CoV-2 infections from patients due to perceived or real deficiencies in PPE, our findings suggest that the majority of HCW infections during the first pandemic wave were acquired from other HCWs. This finding is supported by similar results from a prior simulation study[16]. The contribution of these staff-to-staff infections to hospital-acquired transmission reduced dramatically during the autumn 2020 wave of SARS-CoV-2. Staff were less likely to initiate hospital transmission chains in Wave 2, accounting for 31.3% of index cases compared to 50.6% in Wave 1. Infection control practice and understanding of SARS-CoV-2 transmission evolved considerably

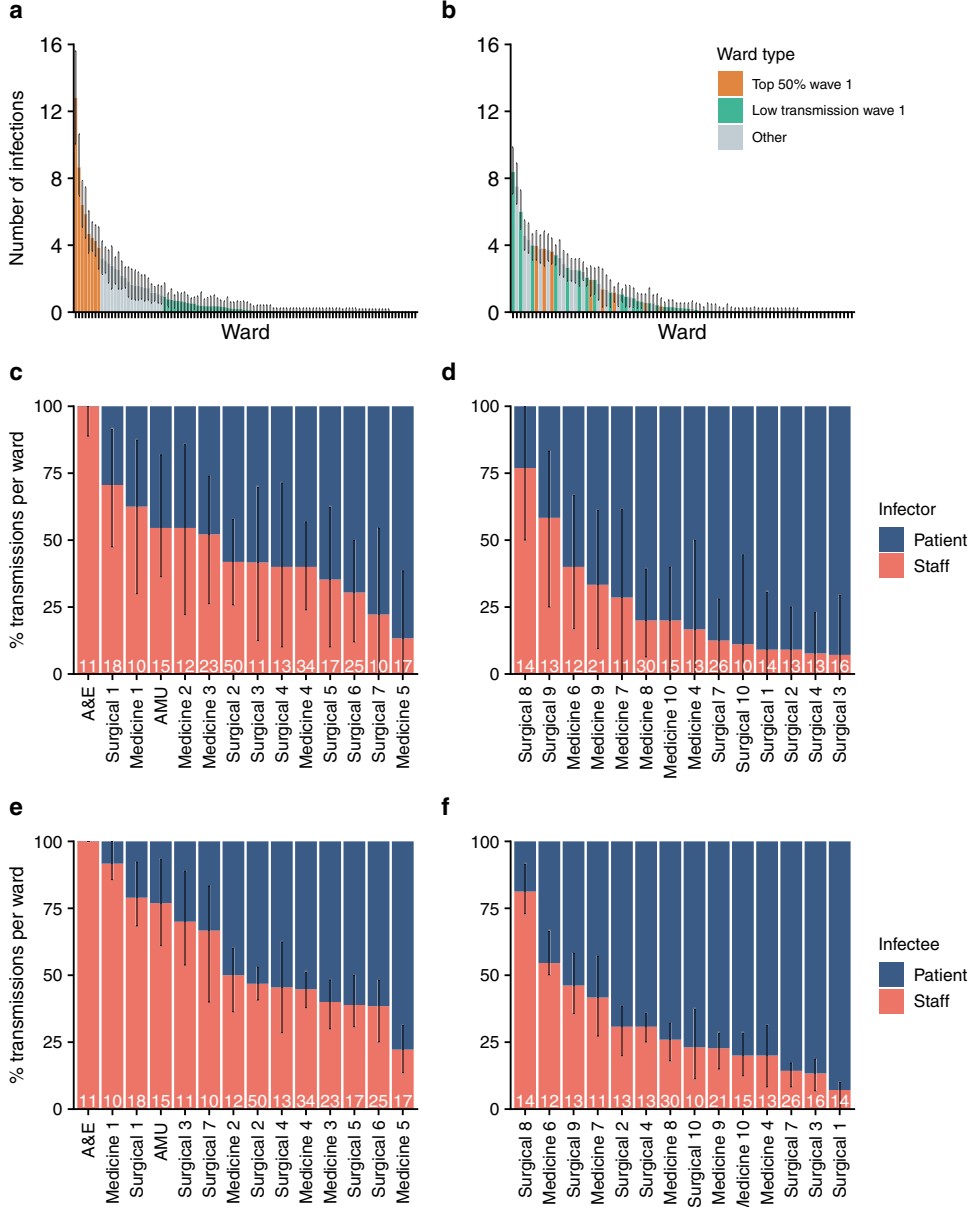

**Fig. 3 Hospital locations contributing to transmission events. a** The number of infections per ward in Wave 1. **b** The number of infections per each ward in Wave 2. Wards contributing to 50% of all transmission events in Wave 1 ($n = 8$) are coloured in orange in both Wave 1 and Wave 2 datasets. Wards with <1 transmission (mean of 10,000 networks) in Wave 1 are coloured in green in both Wave 1 and Wave 2 datasets. **c–f** Percentage of staff and patient infector and infectees per ward involved in transmission events. All wards with 10 or more transmissions in Wave 1 and Wave 2 are shown. Wards are ordered by the percentage of staff infector cases in Wave 1 **c** and Wave 2 **d** and percentage of staff infectee cases in Wave 1 **e** and Wave 2 **f**. Numbers of transmission events per ward are shown at the bottom of each column. The percentage of staff-staff transmissions in A&E is artificially high (100%) due to the inability to obtain patient movement data in this location. Error bars represent 95% credible intervals. A&E accident and emergency department; AMU acute medical unit.

during the pandemic. Improved social distancing and wearing of face coverings in non-clinical areas may explain some of these observations. In addition, the importance of asymptomatic transmission was increasingly appreciated and twice weekly lateral flow testing for healthcare workers was introduced in December 2020. Furthermore, seroprevalence rates of over 25% have been reported in HCWs following the first pandemic wave[17], including in our NHS Trust[18], which may have contributed to greater protection and reduced transmission in some areas. For example, we have reported that the SARS-CoV-2 seroprevalence in staff working on our acute medical unit by June 2020 was over 40%[18]. This was an area that was a hotspot

for transmissions involving staff during our Wave 1 analysis but had very few transmission events identified during Wave 2. Patient to staff transmissions remained constant both in terms of absolute and proportion of transmission events across the two waves, suggesting that whatever factors are responsible for the reduction in staff-staff infections had limited impact on the risk of patient-to-staff transmissions. By Wave 2, most staff infections in our NHS trust were estimated to have been acquired from patients, so further efforts are required to increase protection for HCWs. Staff vaccination is anticipated to have a large impact but is unlikely to have played a significant role in our observations due to the introduction towards the end of Wave 2.

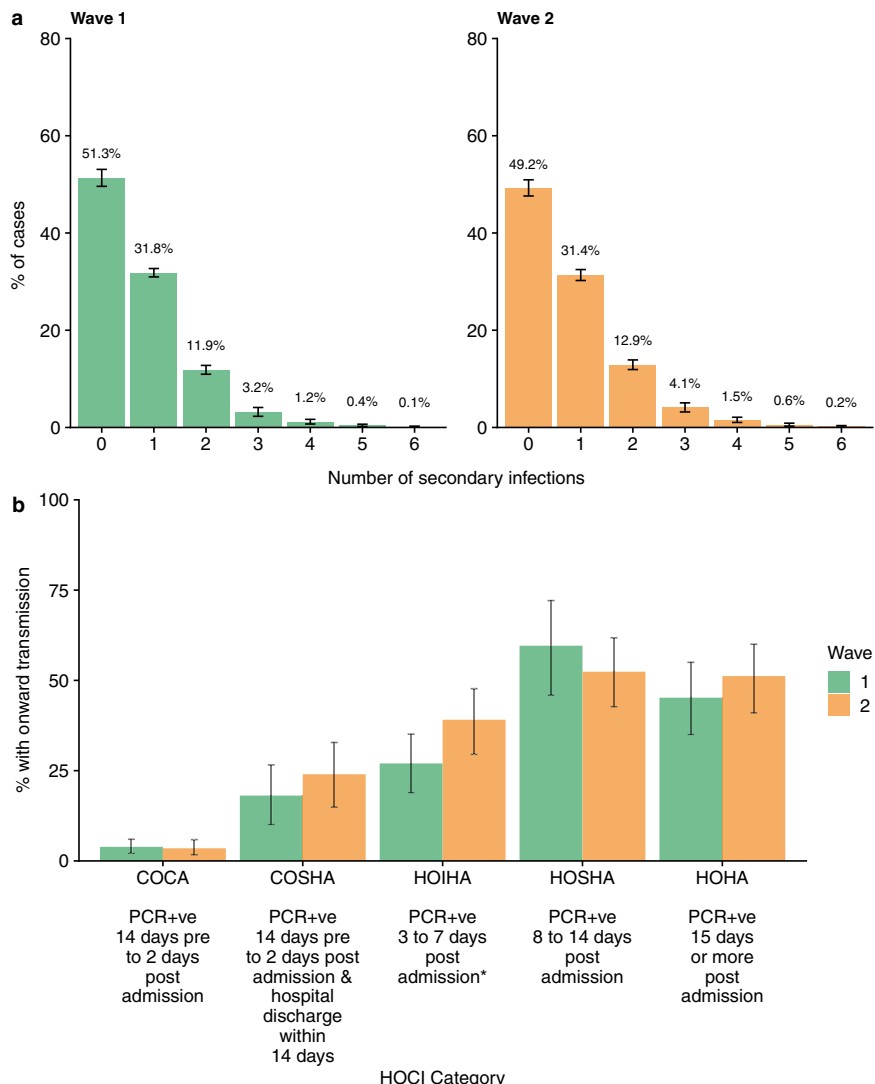

**Fig. 4 Secondary cases distributions and onward infections by Hospital Onset COVID19 Infection category. a** the percentage of cases with each number of secondary cases after adjusting for cases involved in transmissions but not represented in the dataset. **b** The percentage of each HOCI category case with at least 1 onward infection. Error bars represent 95% credible intervals. COCA Community onset community associated; positive test up to 14 days before or within 2 days after hospital admission. COSHA Community onset suspected hospital associated; positive test up to 14 days before or within 2 days after admission, with discharge from hospital within 14 days before test. HOIHA Hospital onset intermediate hospital associated; positive test 3–7 days after hospital admission, (*) with no discharge from hospital in 14 days before the specimen date. HOSHA Hospital onset suspected hospital associated; positive test 8–14 days after admission or 3–14 days after admission with discharge from hospital in 14 days before test. HOHA Hospital onset hospital associated; positive test 15 or more days after hospital admission. Classification of patient cases according to likely source of infection (community or hospital-acquired) is based on SAGE criteria[15]. Number of cases per HOCI category is shown in Table 1.

Hospital-acquired infections during Wave 2 were overwhelmingly dominated by patient-to-patient transmissions. The reasons for these events are likely to be multifactorial. UK hospitals faced significant bed pressures during this period and unlike during Wave 1, attempts were made to maintain as many routine and elective procedures for as long as possible. By this point, all patients in our hospitals were being routinely tested by NAAT for SARS-CoV-2 on admission and on day 5. Accordingly, the percentage of patients included in our dataset with asymptomatic infection increased from 10.4% during Wave 1 to 23.9% in Wave 2. The intense increase in patient-to-patient infections unfortunately occurred despite this enhanced focus on preventing asymptomatic transmission. Most of our transmission hotspots were wards built over two decades ago, with 6–8 beds per bay, and shared toilet facilities between every 1 to 2 bays[19]. While ventilation in these settings is in line with applicable regulations

at the time of construction, none were designed with a respiratory pandemic in mind. Any contribution from these fixed estate issues will be challenging to address in a short timeframe. While viruses with greater transmissibility could also have played a role during Wave 2, circulation of the B.1.1.7/alpha variant occurred relatively late in our region compared to many other parts of the UK, and many Wave 2 transmission events were due to other SARS-CoV-2 lineages. This is in keeping with a recent study demonstrating that B.1.1.7/alpha infections did not result in greater hospital-acquired infections in the UK[20].

Ward design differs throughout the hospital trust with a varying number of side rooms and bed bays. The number of beds in each shared patient bay ranges from 4 to 8. Of the top 10 wards with the most number of transmission pairs, 7 were wards with 6 or more beds per bay. High attack rates have previously been reported between patients in shared occupancy spaces and factors

such as bay size are likely to explain why some hospital wards experienced a greater number of hospital-acquired cases compared with others[21]. Of note, very few transmissions were estimated to have occurred on critical care units, which may have a number of explanations including universal use of enhanced PPE.

We found that the distribution of secondary cases was very similar across both waves, with ~50% of SARS-CoV-2 cases resulting in onward transmission, although only 5–10% of all infections resulted in more than two secondary cases, matching findings from another UK based study[22]. Our findings are different from those in a smaller study focusing on a few large clusters in another UK hospital, where 20% of individuals caused 80% of transmission events[11]. Although there is no clear threshold for the number of cases of a superspreading event, we did not find many examples where a high number of cases were associated with a single case. On average, the maximum number of individuals linked to the same index was six across all networks. Our findings do not support superspreading events forming a significant proportion of all hospital-acquired infections. This may in part be due to our focus on the entire hospital environment rather than on specific epidemiologically identified outbreaks.

Importantly, we find that hospital-acquired SARS-CoV-2 cases give rise to a greater number of secondary cases than community-onset community-associated cases supporting previous findings[23]. Cases admitted from the community already suspected as having COVID-19 will have been isolated in single cubicles or COVID-19 cohort areas more rapidly, thus limiting opportunities for onward transmission. As severe disease requiring hospitalisation often occurs later in infection, they may also be at a less infectious stage, although hospitalised cases may also shed viable virus for longer[24]. In contrast, individuals with no SARS-CoV-2 symptoms who later acquired nosocomial infection may have initially be placed in bays with other susceptible patients, all of whom tested SARS-CoV-2 negative on screening tests at admission. Given the high viral loads during the first few days of infection, including during pre-symptomatic stages[25], our data suggest that these individuals may acquire SARS-CoV-2 in hospital and have ample opportunity for onward transmission before being detected and isolated. This finding indicates that asymptomatic testing of patients on admission and day 5 was insufficient to prevent these scenarios. Daily testing of patients in the first week of admission or more regular testing throughout admission may allow greater opportunity for intervention, as well as more recent recommendations to IPC guidance such as routine wearing of masks by all patients in bays. Equally, rapid point-of-care testing (POCT) on admission may also reduce the window for transmission early in admission as it allows earlier isolation of asymptomatic community-acquired cases. Our Trust instituted POCT for all medical admissions in mid-January 2021.

Our study has several limitations that are important to consider. Firstly, despite the large number of individuals included, this is a single centre study and may not be generalisable across all UK hospitals given the heterogeneity in practice, building infrastructure, and patient population that exists. Our organisation had a high number of documented hospital-acquired infections in patients between March 2020 and March 2021 ($n = 795$), but was not an outlier with 7 other NHS Trusts with higher numbers (highest $n = 1463$)[26]. Seven of the top 10 busiest NHS Trusts (including our own) were also in the top 10 Trusts with the highest number of hospital-acquired COVID-19 infections in patients, indicating a common theme that may be a driver of nosocomial SARS-CoV-2 infections[27]. The effectiveness of various infection control measures on within-hospital transmissions over time in our setting is likely to be generalisable to many UK hospitals, as they were based on national guidance applicable to all NHS Trusts.

Although we did not have a selective sampling strategy, either for case detection or sequencing of positive cases, it is possible that there was an unobserved sampling bias. For example, as individuals with higher viral loads will be more infectious and their samples more likely to result in successful sequencing, they are more likely to have been included in our dataset. As IPC practice evolved during the course of the pandemic, testing of all asymptomatic patients and staff in outbreak wards only commenced towards the latter part of the first wave, but was routine practice along with other new measures during wave 2. This could have had an impact on some comparisons like the size of outbreaks across waves or if more systematic sampling in this way increased detection proportionally in one group (e.g. patients) than another (e.g. staff). Our model attempted to account for cases that were not represented in the dataset but would not mitigate any bias entirely. Some of our key conclusions, such as the greater onward transmission from hospital-acquired cases was also consistently seen across both waves. It is important to consider that staff-to-staff transmission in non-clinical areas (both inside and outside of the clinical setting) can also be an important driver of HCW infections due to social and behavioural factors which are difficult to adequately quantify in models. While we had electronic records of precise location data for patients during all times of their admission, staff location data were less granular and dependent on self-reported areas of work in the 14 days prior to infection. We undertook sensitivity analyses to test several assumptions regarding priors in our model, as outlined in Supplementary Figs. 2–3 but some assumptions remain unexplored. For example, we assumed that the probability of inclusion in our dataset was the same for both staff and patients. Further granularity could be considered in future developments of the outbreaker model.

With this study, we provide evidence that the integration of clinical surveillance data, viral genomic information and modelling enhances our capacity to unravel the complex transmission dynamics of SARS-CoV-2 in times and places of high incidence. The application of such a high-resolution framework to healthcare settings offers attractive perspectives for guiding the development of a safe environment for both staff and patients, as it may have a significant impact on the reduction of SARS-CoV-2 hospital transmission in subsequent epidemic waves.

## Methods

**Study population**. All cases in the study were patients or staff who tested positive for SARS-CoV-2 at Sheffield Teaching Hospitals NHS Foundation Trust (STHNFT), Sheffield, UK, between 1st March 2020 and 24th January 2021. STHNFT is a large UK NHS hospital Trust which includes five hospitals, has an average bed occupancy of 1400, and employs ~17,000 staff. SARS-CoV-2 nucleic acid amplification tests (NAAT) were performed on nose and/or throat swabs throughout the pandemic in line with contemporaneous UK Department of Health and Social Care guidance[28], using Hologic Panther or an in-house dual E/RdRp gene real-time PCR assay[29,30].

Patients were included in the analysis if they tested positive for SARS-CoV-2 at or during admission. Staff were included if they tested positive for SARS-CoV-2 and had worked in a clinical area in the 14 days prior to a positive test. Information on symptom onset of patients and their ward movements, together with place of work for staff, were extracted from STHNFT electronic records, when available.

**Sample Preparation, ARTIC Network PCR and Nanopore Sequencing**.
Sequencing was attempted on all available residual samples collected for routine diagnostic testing from STHNFT throughout the study period, with fluctuation in the proportion of positive samples sequenced due to multiple factors, including laboratory capacity and availability of stored samples. There was no systematic strategy to sequence samples from suspected outbreak wards alone. The first positive sample from each individual was selected for sequencing. RNA was extracted from viral transport medium and subject to the ARTIC network tiled amplicon protocol[31], followed by sequencing on an Oxford Nanopore GridION X5. Base calling was performed using a high accuracy model and the default basecaller in MinKNOW (currently guppy v4). Reads were filtered based on quality and length (400 bp to 700 bp) and mapped to the Wuhan reference genome (GenBank accession number NC_045512). Reads were downsampled to 200x coverage in each direction and variants called using nanopolish[32] to determine changes from the reference, followed by consensus sequence generation. Samples with over 90%

genome coverage were included for further analysis. Viral genomic sequences were classified into PANGO lineages using the *Phylogenetic Assignment of Named Global Outbreak LINeages* (PANGOLIN)[33] version 2.4.2 and a multiple sequence alignment built using MAFFT[34] with 10 iterative refinements. All alignment positions flagged as problematic for phylogenetic inference were removed, including highly homoplasic positions and 3′ and 5′ ends[35].

**Hospital outbreak reconstruction model.** We used Outbreaker2, a modular discrete-time stochastic model for reconstructing likely transmission trees of an outbreak based on pathogen genetic sequences and their collection dates in a bayesian framework via MCMC[12,13]. To investigate nosocomial outbreaks, we extended the most recent implementation of the Outbreaker2 model to capture ward-level transmission by incorporating ward occupancy data and probabilistically favouring infections that occurred within a ward rather than between individuals on different wards[36]. Our Bayesian model calculates the likelihood of a transmission event from case i to case j at a putative transmission time, given the time of symptom onset for case i and j, the Hamming distance between the corresponding virus genetic sequences, and the ward that i and j were on at the time of infection. The model also infers unobserved infections and unobserved transmission pathways using a constant reporting rate parameter (an ascertainment probability, i.e., the proportion of all SARS-CoV-2 positive cases in admitted patients and hospital staff that were captured in our dataset). This parameterization (outlined in Supplementary Methods) allows to infer unobserved transmission pathways linking a given ward to another given ward over consecutive generations of infection.

We estimated the ascertainment probability as the product of (i) the proportion of all cases that were likely detected via testing, (ii) the proportion of detected cases with high-quality sequence, and (ii) the proportion of these cases where the ward location was known. We used a point estimate of 0.5 for this ascertainment probability but varied this estimate in a one-way sensitivity analysis (Estimates of 0.3, 0.4, 0.6 and 0.7; Full details in Supplementary Methods).

We also extended this model to estimate the number and identity of imported community-acquired infections. To do so, we first estimated the fraction of community-acquired infections across the whole dataset using admission dates, symptom onset dates and the incubation period. We then classified cases as either an imported infection or a hospital-associated infection based on the likelihood of a given case being infected by another individual observed in the dataset. The generation time distribution (the delay between the infection of a primary case and the infection of a secondary case), and the incubation period distribution (the delay between the infection and symptom onset) used to inform the inference were based on previously published estimates which incorporate uncertainty around these values[37,38].

In our base-case analysis, we used a global sensitivity method, incorporating all results from a sensitivity analysis in the final results output, to capture uncertainty in (i) the proportion of cases that were community-acquired infections imported into the hospital (Wave 1: N($\mu = 0.7$, $\sigma = 0.075$), Wave 2: N($\mu = 0.6$, $\sigma = 0.06$)), (ii) the symptom onset date for individuals for whom this information was unavailable and (imputed 100 times), and (iii) the place of work for some staff who had multiple work locations (imputed 100 times). A final posterior distribution of 10,000 transmission networks was inferred by integrating over the uncertainty across imputed datasets of these three aspects. Full details of the model, including model fitting and prior distributions, are provided in Supplementary Table 1. All analysis was carried out in R (Version 4.0.3)[39].

**Disclaimer**. The views expressed are those of the author(s) and not necessarily those of the NIHR, Public Health England or the Department of Health and Social Care.

## Data availability
Viral genomes were mapped to the publicly available Wuhan reference genome (GenBank accession number NC_045512). All sequences used in this study are deposited in the European Nucleotide Archive (see Supplementary data 1 for accession numbers). The epidemiological data and linkage to sequences are available under restricted access due to their potentially identifiable nature. Access can be obtained by contacting the corresponding author (t.desilva@sheffield.ac.uk) after which a data sharing agreement will be organised. We will aim to respond to any requests within 10 working days.

## Code availability
All code is available at https://github.com/Chjulian/sheffield_HT[40].

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

## Acknowledgements

We thank the Sheffield Bioinformatics Core for their thoughtful discussion. We would like to thank the members of the Sheffield Biomedical Research centre for their continued support of the SARS-CoV-2 sequencing work in Sheffield. We thank all partners of and contributors to the COG-UK consortium, who are listed at https://www.cogconsortium.uk/about/. Sequencing of SARS-CoV-2 samples was undertaken by the Sheffield COVID-19 Genomics Group as part of the COG-UK CONSORTIUM and supported by funding from the Medical Research Council (MRC) part of UK Research & Innovation (UKRI), the National Institute of Health Research (NIHR) and Genome Research Limited, operating as the Wellcome Sanger Institute. M.D.P. and D.W. are funded by the NIHR Sheffield Biomedical Research Centre (BRC - IS-BRC-1215-20017). T.I.dS. is supported by a Wellcome Trust Intermediate Clinical Fellowship (110058/Z/15/Z). C.J.V.A. and K.E.A. were funded by an ERC Starting Grant (action number 757688). This study is partially funded by the NIHR Health Protection Research Unit in Modelling and Health Economics, a partnership between Public Health England, Imperial College London and LSHTM (grant code NIHR200908); and acknowledges funding from the MRC Centre for Global Infectious Disease Analysis (reference MR/R015600/1), jointly funded by the UK MRC and the UK Foreign, Commonwealth & Development Office (FCDO), under the MRC/FCDO Concordat agreement and is also part of the EDCTP2 programme supported by the European Union.

## Author contributions

Contributor roles were assigned as per http://credit.niso.org/. K.E.A., S. Hué and T.I.dS. contributed equally. B.B.L., C.J.V.A., F.C., K.E.A., S.H. & T.I.dS. were involved in the conceptualisation of the study. B.B.L., A.J.K., M.D.P., D.R.S., P.Z., N.K., M.G., B.H.F., P.W., S.F.L., S.C., A.S., K.J., M.R., S. Hsu, H.P., C.M.E., D.G.P., K.E.A., S. Hué and T.I.dS. were involved in data collection and curation. B.B.L., C.J.V.A., F.C., M.D.P., S. Hué and K.E.A. were involved in data analysis. A.C., T.J., S. Hué, K.E.A. and T.I.dS. were involved in the supervision of the project. B.B.L. and C.J.V.A. were involved in data visualisation. F.C. was involved in software development. B.B.L., C.J.V.A., F.C., K.E.A., S. Hué & T.I.dS. wrote the original draft. All authors were involved in reviewing and editing the final manuscript. Members of the Sheffield COVID-19 Genomics Group contributed to the generation of the sequence data used. Members of the CMMID COVID-19 Working Group contributed to the interpretation of data and approved the work for publication following manuscript review. Members of the COVID-19 Genomics UK (COG-UK) consortium contributed to data curation and analysis.

## Ethical approval

Approval for the study was obtained from the UK Health Research Authority (IRAS 281918), with sequencing performed according to The COVID-19 Genomics UK (COG-UK) study protocol approved by the Public Health England Research Ethics Governance Group (R&D NR0195). Approval was provided to undertake viral sequencing on residual clinical diagnostic samples and analysis on pseudo-anonymised data without individual patient consent.

## Competing interests

The authors declare no competing interests.

## Additional information

## Sheffield COVID-19 Genomics Group

Benjamin B. Lindsey [1,2,15], Alexander J. Keeley [1,2], Matthew D. Parker [6,7,8], Dhruv R. Shah[1,2], Helena Parsons[1,2], Peijun Zhang[1], Marta Gallis[1], Benjamin H. Foulkes [1], Paige Wolverson [1], Stavroula F. Louka[1], Stella Christou[1], Amy State[2], Katie Johnson[1,2], Mohammad Raza[1,2], Sharon Hsu[1], Cariad M. Evans[1,2], David G. Partridge [1,2], Thushan I. de Silva[1,2], Alison Cope[1,2], Nasar Ali[2], Rasha Raghei[2], Joe Heffer[12], Nikki Smith[1], Max Whiteley[1], Manoj Pohare[1], Samantha E. Hansford[1], Luke R. Green[1], Dennis Wang[6,7,8,13], Michael Anckorn[1,2], Adrienn Angyal[1], Rebecca Brown[1], Hailey Hornsby[1], Mehmet Yavuz[2],

Danielle C. Groves[1], Paul J. Parsons[14], Rachel M. Tucker[14], Magdalena B. Dabrowska[1,7], Thomas Saville[1], Jose Schutter[1] & Matthew D. Wyles[8]

[12]IT services, The University of Sheffield, Sheffield, UK. [13]Department of Computer Science, The University of Sheffield, Sheffield, UK. [14]Department of Animal and Plant Sciences, The University of Sheffield, Sheffield, UK.

## The COVID-19 Genomics UK (COG-UK) consortium

Matthew D. Parker [6,7,8], Thushan I. de Silva[1,2], Sharon Hsu [1,7], Peijun Zhang[1], Marta Gallis[1], Stavroula F. Louka[1], Benjamin B. Lindsey [1,2,15], Alexander J. Keeley [1,2], David G. Partridge [1,2], Mohammad Raza[1,2] & Cariad Evans[1,2]

## CMMID COVID-19 working group

Ch. Julián Villabona-Arenas [3,4,15], Thibaut Jombart[3,4,9], Nicholas G. Davies[3,4], Carl A. B. Pearson[3], Matthew Quaife[3], Damien C. Tully[3,4], Sam Abbott[3,4], Katherine E. Atkins [3,4,10✉] & Stéphane Hué [3,4✉]

A full list of members and their affiliations appears in the Supplementary Information.

