## [Peer Review File · Nature Communications]

Characterising within-hospital SARS-CoV-2 transmission events using epidemiological and viral genomic data across two pandemic wavesEditorial Note: This manuscript has been previously reviewed at another journal that is not operating a transparent peer review scheme. This document only contains reviewer comments and rebuttal letters for versions considered at *Nature Communications*.

REVIEWERS' COMMENTS

Reviewer #1 (Remarks to the Author):

For reference, my prior comments are listed on the response memo under reviewer 1. The authors have fully addressed my concerns and have appropriately framed the limitations I previously highlighted.

To briefly summarize, then, this is an interesting and well-conducted analysis that has been significantly strengthened by these revisions. It has important implications for understanding in-hospital transmission dynamics, and uses a relatively novel approach for the hospital setting that combines epidemiologic and genomic data. The importance of staff-to-staff transmission, nosocomial > community acquired cases driving secondary cases, and preponderance of transmission within certain wards with high numbers of beds within bays are critical insights that can guide improvements in infection control practices, particularly in the setting of the more-transmissible Delta variant.

One point of interest since my last review, in a similar analysis of the outbreak in Provincetown, Massachusetts (<https://doi.org/10.1101/2021.10.20.21265137>), those investigators also used outbreaker2 and were able to incorporate use of iSNVs to infer directionality of transmission. While I don't think this necessarily needs to be included in this paper, it may be interesting to look at in subsequent analyses to better understand transmission between staff and patients, for example.

I congratulate the authors on this work and recommend accepting this manuscript for publication.

Aaron Richterman

Reviewer #3 (Remarks to the Author):

Many thanks for the opportunity to review this revised manuscript, and many thanks to the other journal and Nature for allowing the transfer of my previous comments. A sensible approach that saves time.

I still think this manuscript is very good and is an important contribution to the literature. The responses to my comments have been adequately addressed. I have little to add. There is an error in referencing on line 456.

I have no other minor comments. Congratulations to the authors for this piece of work.

Reviewer #4 (Remarks to the Author):

All my comments on a previous version of this manuscript (reviewed for the version submitted to another journal) have now been appropriately addressed. The only remaining changes that I think are needed are to address a few typos and minor notational issues in the SI:

p2 typo: "and and infecteé "

p4 "for example as given by proximity *or* the number of staff shared between them"

p5 I am happy with the the definition of $X(\sigma)$ at the top of p5 in the sense that I can see what

all the elements are and they make sense, but the notation for defining the matrix (with no separators between matrix elements or lines) is new to me and seems unnecessarily confusing (and the new text in fact makes the first line of page 5 redundant). Of course, this could also just be a Word formatting issue!

p6 " as these infection events are not linked *to* either of the observed cases"

p9 The equation at top of this page seems to have a couple of typos - please check (could also be a Word formatting issues again).

p13 typo: "We first calculated, for each patient, i , the probability that they acquired we observed a delay between ..."

REVIEWERS' COMMENTS

Reviewer #1 (Remarks to the Author):

For reference, my prior comments are listed on the response memo under reviewer 1. The authors have fully addressed my concerns and have appropriately framed the limitations I previously highlighted.

To briefly summarize, then, this is an interesting and well-conducted analysis that has been significantly strengthened by these revisions. It has important implications for understanding in-hospital transmission dynamics, and uses a relatively novel approach for the hospital setting that combines epidemiologic and genomic data. The importance of staff-to-staff transmission, nosocomial > community acquired cases driving secondary cases, and preponderance of transmission within certain wards with high numbers of beds within bays are critical insights that can guide improvements in infection control practices, particularly in the setting of the more-transmissible Delta variant.

One point of interest since my last review, in a similar analysis of the outbreak in Provincetown, Massachusetts (<https://doi.org/10.1101/2021.10.20.21265137>), those investigators also used outbreaker2 and were able to incorporate use of iSNVs to infer directionality of transmission. While I don't think this necessarily needs to be included in this paper, it may be interesting to look at in subsequent analyses to better understand transmission between staff and patients, for example.

I congratulate the authors on this work and recommend accepting this manuscript for publication.

Aaron Richterman

Response: Thank you for your kind comments and for taking the time to review our paper. We will review the recommended paper and consider incorporating it into our methods in the future.

Reviewer #3 (Remarks to the Author):

Many thanks for the opportunity to review this revised manuscript, and many thanks to the other journal and Nature for allowing the transfer of my previous comments. A sensible approach that saves time.

I still think this manuscript is very good and is an important contribution to the literature. The responses to my comments have been adequately addressed. I have little to add. There is an error in referencing on line 456.

I have no other minor comments. Congratulations to the authors for this piece of work.

Response: Thank you for the kind comments. We have fixed the referencing error highlighted.

Reviewer #4 (Remarks to the Author):

All my comments on a previous version of this manuscript (reviewed for the version submitted to

another journal) have now been appropriately addressed. The only remaining changes that I think are needed are to address a few typos and minor notational issues in the SI:

p2 typo: “and and infecteé “

p4 “for example as given by proximity *or* the number of staff shared between them”

p5 I am happy with the the definition of $X(\sigma)$ at the top of p5 in the sense that I can see what all the elements are and they make sense, but the notation for defining the matrix (with no separators between matrix elements or lines) is new to me and seems unnecessarily confusing (and the new text in fact makes the first line of page 5 redundant). Of course, this could also just be a Word formatting issue!

p6 “ as these infection events are not linked *to* either of the observed cases”

p9 The equation at top of this page seems to have a couple of typos - please check (could also be a Word formatting issues again).

p13 typo: “We first calculated, for each patient, i , the probability that they acquired we observed a delay between ...”

Response: Thank you for the thorough review of our paper and spotting these typos/formatting errors. We’ve now corrected all of these.